# Compliance among Registered Nurses and Doctors in Critical Care Units: Challenges Affecting Their Adherence to Standard Precautions

**DOI:** 10.3390/healthcare11222975

**Published:** 2023-11-16

**Authors:** Naglaa Abdelaziz Mahmoud Elseesy, Ahlam Eidah Al-Zahrani, Faten Shawky Kandil, Alaa Mahsoon, Mona Mohamed Elhady

**Affiliations:** 1Nursing Administration Department, Faculty of Nursing, Alexandria University, Alexandria 21527, Egypt; naalsesei@kau.edu.sa; 2Public Health Department, Faculty of Nursing, King Abdulaziz University, Jeddah 21589, Saudi Arabia; 3Maternity and Childhood Nursing Department, Faculty of Nursing, King Abdulaziz University, Jeddah 21589, Saudi Arabia; aealzahrani@kau.edu.sa; 4Nursing Department, Faculty of applied Medical Sciences, University of Jeddah, Jeddah 21589, Saudi Arabia; fskandil@uj.edu.sa; 5Medical-Surgical Nursing Department, Faculty of Nursing, Cairo University, Cairo 12613, Egypt; 6Psychiatric and Mental Health Department, Faculty of Nursing, King Abdulaziz University, Jeddah 21589, Saudi Arabia; 7Emergency and Critical Care Nursing Department, Faculty of Nursing, Mansoura University, Elmansoura 35516, Egypt; monamelhady@gmail.com; 8Critical Care Nursing Department, Faculty of Nursing, King Abdulaziz University, Jeddah 21589, Saudi Arabia

**Keywords:** compliance, healthcare workers, standard precautions, critical care units

## Abstract

(1) Background: Awareness and compliance with standard precautions (SPs) are essential health issues for both healthcare staff and patients. Hence, more research is urgently needed in the health sector worldwide, particularly in Arab countries. The purpose of this study was to investigate compliance with SPs among nurses and doctors, as well as challenges affecting their adherence to these SPs. (2) Methods: A cross-sectional descriptive study was conducted. The researchers administered the compliance with the Standard Precautions Scale (CAPS) to staff nurses and doctors from critical care units at a university-affiliated hospital in Jeddah, Kingdom of Saudi Arabia (KSA). A total of 112 nurses and 59 doctors were enrolled in the study to determine their compliance with SPs as well as the challenges affecting their adherence to these SPs. (3) Results: The overall mean percentage scores of nurses’ and doctors’ compliance with SPs were 36.43 ± 15.85% and 33.27 ± 15.76%, respectively. In addition, the current study identified associated challenges affecting these healthcare workers’ adherence to these SPs, including the effect of sociodemographic factors, such as age and experience. (4) Conclusions: The study found that both nurses and doctors had poor compliance with all SPs except for disposal of sharps, which scored with suboptimal compliance. Nurses’ compliance with recommended precautions varied significantly according to age, clinical experience, and attendance at training courses. Healthcare organizations should consider a holistic approach to promote adherence to SPs to enhance outcomes associated with optimized care delivery and stakeholder safety.

## 1. Introduction

Despite healthcare-associated infections (HAIs) being preventable, poor compliance with infection prevention and control measures has been a major problem for hospitals [1]. To prevent HAIs and safeguard patients and medical workers from infectious organisms, knowledge of and adherence to established precautions are necessary [2]. In hospital settings, standard precautions (SPs) help prevent the transmission of infections. SPs are specific measures and protocols designed for healthcare practice contexts to help minimize the spread of infectious diseases or contamination through blood, body fluids, excreta, skin, or mucosa, irrespective of whether there is a known or suspected infection [3]. The Centers for Disease Control and Prevention (CDC) defined infection control SPs as “the minimal infection prevention procedures that should be applied to all patient care” [4]. This definition applies regardless of whether the patients’ infection status is suspected or confirmed, and it can be applied in any setting where medical care is provided. Hand hygiene, the use of gloves and other personal protective equipment (PPE), proper cleaning and disinfection of patient care equipment and environment surfaces, proper waste disposal, proper management of used needles and other sharp objects, and proper cough etiquette are all part of the SPs guidelines [5]. SP integration into healthcare delivery models helps preserve the health and well-being of patients, healthcare providers, and other stakeholders during care provision while optimizing the relative success of associated interventions. Nevertheless, adherence to SPs has been established as an inherent challenge for most healthcare institutions, with adherence rates ranging from a concerning 19.5% to a more favorable above 80% in some studies [6].

Non-compliance with infection control guidelines increases the “risk of contracting and spreading infectious diseases among healthcare workers (HCWs)” [7]. Poor infection control can result in HAIs; therefore, compliance with SPs in critical care units is the primary strategy for preventing HAIs [5,7]. However, the HCWs’ adherence to SPs was found to be insufficient [8]. In the Kingdom of Saudi Arabia (KSA), non-compliance with SPs is the major source of medical errors and poor patient outcomes [9]. Given the notable risks and potential adverse implications to healthcare delivery effectiveness, non-adherence has emerged as a critical concern for practitioners and scholars alike, necessitating a critical review of the phenomenon to help understand causal factors and feasible solutions to enhance adherence. Subsequently, several factors have been highlighted within the diverse body of literature. For instance, Bouchoucha and Moore (2019) identified individual judgment, leadership, culture/practice, contextual cues, and justifications as the integral factors influencing adherence to SPs [6]. These have been additionally corroborated by Mendes (2019) and Nwankwo et al. (2020), identifying non-adherence to be a complex product of individual, organizational, and structural constraints [10,11]. A possible reason for non-compliance with SPs among HCWs is the lack of knowledge [12]. This creates a perfect storm [13] in which norms may be violated and not adhered to, enforced, or considered necessary. This has resulted in increased medical costs. As a result of these errors, patient death rates are increasing [14]. Indeed, medical error has become the third leading cause of death in the United States due to the magnitude of this problem, which has been exacerbated by the present COVID-19 pandemic and the heightened demand for infection prevention. This area of research requires further investigation [15]. Despite SPs being prominent for a long time, incomplete compliance to SPs by HCW remains a concern for many organizations, particularly those in industrialized countries [16]. Non-compliance to SPs is frequently associated with insufficient knowledge of SPs, lack of PPE, poor perception of risk, and low perception of the institutional safety environment [17]. SPs require compliance by both doctors and nurses since they are partners in improving the quality of patient care. If either group does not comply with SPs, the standard of patient care in all areas of hospital settings can suffer [18,19].

Self-reported compliance with SPs among Dental Health Care Workers (DHCWs) is reportedly high [20], although this requires confirmation in a follow-up direct observation study. Haridi, Al-Ammar, and Al-Mansour (2016) also emphasized the necessity of infection control education in promoting compliance with SPs, as well as the need to engage dental assistants and DHCWs in the commercial sector in particular [20]. Bahegwa et al. (2022) acknowledge the complex interplay of factors within the healthcare context, arguing that enhanced adherence should result from practitioner training to enhance competence and capacity, existing support or supervision, adaptability within the work context, and experience [21]. In addition, more research is urgently needed in the KSA and other Arab countries on the awareness and compliance with SPs as one of the most essential health issues for both HCWs and patients.

CDC recommends SPs for preventing the transmission of infections during patient care. However, several challenges affect the knowledge of and compliance to SPs [2]. Therefore, the goal of this research was to assess the present compliance rates with SPs among nurses and doctors in critical care units, and to investigate the challenges that might be linked to adherence to SPs in practice.

## 2. Materials and Methods

### 2.1. Study Design, Setting

A cross-sectional survey was conducted in critical care units at University Hospital in Jeddah, KSA. Three critical care units were included: a medical intensive care unit, a surgical intensive care unit, and a coronary care unit. Critical care units were selected since they treat critical cases of both male and female patients in various fields, including maternity, obstetrics, medical, and surgical. In addition, they require the implementation of SPs by HCWs.

### 2.2. Sampling

Convenience (nonprobability) sampling was used to obtain the study sample. Convenience sampling was chosen due to the descriptive nature of the study. A convenience sample of 161 participants comprising nurses (*n* = 112) and doctors (*n* = 49) were recruited. Based on the extant literature on infection control and prevention standards [22], using the following statistical parameters: medium effect size of 0.30, alpha of 0.10, and power of 0.80, a total of 161 subjects is sufficient to ensure statistically and clinically meaningful results.

### 2.3. Ethical Considerations

Official approval was obtained from the Faculty of Nursing at King Abdulaziz University (Ref No 2F.76). The privacy, anonymity, and confidentiality of the data were maintained and assured by obtaining participants’ agreement to participate in the research before data collection. Participants had the right to withdraw from the study at any time.

### 2.4. Study Tools

A structured questionnaire was used to collect the data. The questionnaire consisted of three parts, as follows:*Sociodemographic characteristics*

Questions on the sociodemographic characteristics (e.g., age, gender, marital status, nationality, experience, and training course) of the participants were included.


*Challenges affecting healthcare workers’ practice of standard precautions*


Challenges affecting HCWs in their performance of standard precautions were developed by Ogoina et al. [23] and utilized by the researchers who collected data to measure challenging factors affecting HCWs’ adherence to SPs. Each statement was presented on a measured scale using a dichotomous yes or no answer. The respondents were each asked to rate the challenges affecting their adherence to SPs.


*Compliance with the Standard Precautions Scale*


The Compliance with the Standard Precautions Scale (CSPS) was developed by Lam [18]. It comprises a 20-item checklist that covers topics such as the use of PPE, disposal of sharp instruments and waste, decontamination of spills and used items, and avoidance of cross-infection. The response set was a 4-point Likert scale with responses and scores comprising “never = 1”, “rarely = 2”, “sometimes = 3”, and “always = 4”. The total scores ranged from 0 to 20, with a higher number indicating a higher level of adherence to SPs.

The CSPS content and construct validity was verified by Lam [24]. The tool’s reliability was also assessed by Cronbach’s α coefficient, which was 0.875 and deemed acceptable for the current study sample.

### 2.5. Data Collection

Data were collected from 1 February to May 2019. The questionnaire was hand-delivered to staff nurses and doctors in their work setting after an individualized interview with each nurse and doctor that lasted about 2 min to explain the goal of the study; they were informed of the information that their participation was voluntary, and no personal or identification information would be required. Informed consent was waived due to the anonymity of the participants and was implied by the participants returning the filled questionnaire to the researchers collecting the data. The estimated time to fill out the questionnaires was 15–20 min.

### 2.6. Data Analysis

The data were analyzed using SPSS software (version 20.0). Quantitative data were presented with numbers and percentages, means, and standard deviations. The level of significance for the obtained results was determined to be 0.05. The Chi-squared (χ^2^) test for categorical variables was used to compare different groups. Monte Carlo correction was used for χ^2^ when more than 20% of the cells had an expected count of less than 5. The two studied groups were compared using an independent sample *t*-test for normally distributed quantitative variables. Moreover, an *F*-test (analysis of variance) was adopted for normally distributed quantitative variables to compare between more than two groups, and a post hoc test (Tukey’s using least significant difference) was used for pairwise comparisons. The internal consistency of the scale was determined through Cronbach’s α coefficient.

## 3. Results

### 3.1. Participants’ Demographic Characteristics

The demographic characteristics of the study participants are shown in Table 1. A total of 161 HCWs participated in the study; 112 (69.57%) were nurses, 49 (30.43%) were doctors, and the overall response rate was 100%. Nurse respondents mainly featured females with a mean age of 32.50, with a diploma as the highest education level, married, and non-Saudi. On the other hand, the doctor respondents mainly featured males, with a mean age of 39.71, with a Doctoral degree, married, and of Saudi nationality. The doctors were much older than the nurses but had similar clinical experience (i.e., 5–14 years). As for nationality, 65.2% of the nurses were non-Saudi, whereas 61.2% of the doctors were Saudi.

Nevertheless, most nurses (97%) and doctors (91%) conceded not having any training in infection control. Moreover, only 58.9% of nurses and 55.1% of doctors stated the need to enroll in training courses.

### 3.2. Total Scores of CSPS between the Doctors and Nurses

Nurses recorded a higher CSPS than doctors, with overall mean percentage scores of nurses and doctors’ CSPS being 36.43 ± 15.85% and 33.27 ± 15.76%, respectively. However, the difference was not significant between the two groups (*p* = 0.245). The most significant difference was in the disposal of waste material. On pairwise comparisons, nurses had a significantly higher mean percentage compliance score than doctors (t = 2.026, *p* = 0.045). The differences were not statistically significant in the other dimensions (Table 2).

### 3.3. Levels of CSPS between the Doctors and Nurses

Across professional groups, the nurses had a poorer overall level of compliance with SPs (78.6%) compared with doctors (85.7%). On evaluating the degree of compliance between doctors and nurses on the different measures for standard precaution, it was deemed that the only significant difference (t = 7.266 *, *p* = 0.041) was that nurses performed better in preventing cross-infection from person to person (mostly poor = 76.8%) and (suboptimal = 20.3%), but with lower percentages than doctors (mostly poor = 87.8%). Additionally, a notable observation from the performance levels analyzed was that both study groups scored poorly on all categories aside from the disposal of sharps, where the best scores were for suboptimal performance (Table 3).

### 3.4. Challenges Affecting Adherence to Standard Precautions

Table 4 shows doctors’ and nurses’ challenges affecting their adherence to standard precautions. The observed differences in reported challenges among doctors and nurses were not statistically significant (*p* > 0.05, all analyses, χ^2^). However, the only significant difference was perception regarding the lack of resources (χ^2^ = 4.076 *, *p* = 0.043) and hospital policy (χ^2^ = 4.068 *, *p* = 0.044) on SPs.

The challenges reported that affected nurses’ adherence to SPs more than doctors were excess workload (82.1%), absence of regular training on infection control (75%), shortage of staff (74.1), lack of hospital policies for practice of SPs (73.2%), lack of adequate facilities and resources for practice of SP (71.4%), lack of a functional infection control committee (69.6%), lack of a standardized documentation tool for standard precautions (67%), lack of management support (63.4%), and lack of knowledge of SP (30%).

In addition, the current study found that “time constraints”, “PPE is uncomfortable”, “patients feel stigmatized when PPE is used”, and “belief that you will not acquire infection in the hospital” were more highly perceived by doctors than nurses as challenges in their adherence to SP practices (9.6%, 71.4%, 53.1%, and 44.9%, respectively).

### 3.5. Standard Precautions and Nurses’ Demographic Data

The findings in Table 5 show that among the 112 nurses, there was a significant correlation between the overall compliance score with SPs and a nurse’s age (f (*p*) 6.075 * (0.003 *)), clinical experience (f (*p*) 5.537 * (0.011 *)), attendance of training courses (t (*p*) 11.138 * (0.001 *)), and need to attend training courses (t (*p*) 2.806 * (0.006 *)). Regarding nurses’ ages, nurses aged from 40 to 49 years had significantly higher mean compliance scores than other age groups (50.83 ± 19.75). As for the clinical experience, highly experienced nurses with more than 20 years had higher overall mean compliance scores (54.29 ± 23.17) than less experienced nurses (41.18 ± 23.02). Highly experienced nurses had higher scores in both their compliance score regarding the use of PPE, decontamination of spills and used items, and prevention of cross infection from person to person. However, less experienced nurses had higher compliance scores regarding waste disposal than more experienced nurses.

Overall, study participants who completed infection control training courses had significantly higher mean compliance scores (36.88 ± 15.82) than those who did not have any such training (20.0 ± 0.0), Furthermore, those who had attended infection control training courses had higher compliance scores in “decontamination of spills and used items” and “prevention of cross infection from person to person”. Similarly, those who wanted to enroll in training courses had a significantly higher mean compliance score (39.55 ± 18.22) than those who did not want to enroll in such courses (31.96 ± 10.25).

Moreover, nurses who wanted to enroll in training had significantly higher mean compliance score regarding use of protective devices and decontamination of spills and used items (34.09 ± 23.28), (19.19 ± 28.68) than those who did not want to enroll in these training courses (26.45 ± 14.31), (9.42 ± 16.72), respectively. There were no significant correlations when compliance score was compared with gender, educational level, marital status, and nationality (*p* > 0.05).

### 3.6. Standard Precautions and Doctors’ Demographic Data

This study found that there was no significant relation between overall compliance score with standard precautions the doctors’ demographics data; gender, age, educational level, marital status, nationality clinical experience, attendance of training courses as well as need to attend training courses. There was significant relation between doctors’ age and compliance regarding sharp disposal (f (*p*) 3.186 * (0.033 *)), where doctors aged less than thirties had significantly higher mean compliance scores than other categories (88.89 ± 16.67) (Table 6).

## 4. Discussion

The aim of this study was to investigate compliance with SPs among nurses and doctors as well as to identify challenges affecting their adherence to SPs. Compliance with SPs is important to prevent HAIs and to protect patients and HCWs from exposure to microbial infections. The present study investigated the statistically significant differences between nurses and doctors in various dimensions of SPs (use of PPE, disposal of sharps and waste, decontamination of spills and used items, and preventing transmission of infections).

The results of this study indicate that more than one-third of both nurses and doctors comply with SPs, with slightly higher compliance scores among nurses than doctors. This result can be attributable to increased non-attendance by both nurses and doctors at SP training programs. Jawaid et al. [25] documented compliance (in percentage) among doctors (81.7%) compared to nurses (97.5%). On the other hand, Ogoina et al. [23] stated that medical staff had significantly higher SP compliance scores than nursing staff. In addition, Eskander et al. [26] concluded that 57.1% of nurses and 57.5% of doctors complied with SPs in Egypt.

HAIs can be limited through HCWs’ compliance with SPs. Nevertheless, the usefulness of SPs in clinical areas is limited due to reduced compliance rates among medical and nursing staff worldwide [2,27]. In the current study, the participants’ compliance with SPs were measured using a 4-point Likert scale. It was observed that the lowest rating (poor) was the highest overall levels of compliance with SPs, which was increased among doctors more than nurses. This result can be attributed to lack of adequate facilities/resources for performance of SPs among doctor and nurses. This result was matched with Haile et al. [8], who found very low total compliance among HCWs (12%).

In addition, these findings were consistent with McGaw et al. [28], which reported low compliance with face mask and uniform requirements. Similarly, Efstathiou et al. [29] revealed low compliance with SPs among HCWs. In addition, Brevidelli and Cianciarullo [30], Primo et al. [31], and La-Rotta et al. [32] revealed poor compliance by doctors with SPs. On the other hand, Al-Faouri et al. [2] reported an intermediate compliance level among studied nurses. Moreover, Carvalho et al. [33] reported intermediate compliance among gynecological doctors.

The current study illustrates how both nurses and doctors rated poorly in their compliance with all SP dimensions apart from disposal of sharps, which was rated suboptimal. This result can be rationalized by understanding the perceived belief of HCWs that they can acquire HAI from needlestick injury, which thus elevates their compliance with the requirement to legally dispose of sharps. However, this result is contradicted by Alice et al. [34], Garus-Pakowska et al. [35], and Nugmanova et al. [36], who concluded that compliance with SPs by HCWs is generally still insufficient.

Conversely, Haile et al. [8] and Alice et al. [34] reported that a relatively higher proportion of HCWs were continuously compliant with SPs practices. Moreover, Ndu and Arinze-Onyia [37] recorded higher compliance by doctors with SP safety guidelines compared with other HCWs. On the other hand, Maheshwari et al. [38] reported higher compliance with handwashing among nursing staff (62.5%) than doctors (21.3%).

Thirteen challenges that can affect compliance by nurses and doctors with SPs were investigated in the current study. The results indicate that insufficient facilities/resources for the practice of SPs and a lack of hospital policy regarding standard precautions were significantly highly reported as preventable challenges from nurses’ perception more than doctors’ perception. This result may have occurred because nursing procedures were performed more frequently than medical procedures that need more facilities. Similarly, Refaei et al. [39] found that insufficient gloves and gowns and difficulty with SP practices were the highest perceived barriers for the nurses’ and doctors’ compliance with SPs.

The results of current study showed that excess workload, absence of frequent training on infection control, staff shortages, lack of hospital policies for the practice of SPs, lack of adequate facilities for SP practice, inefficient infection control panel, lack of a standardized documentation tool for SPs, lack of management support, and lack of knowledge of SPs were reported as challenges that prevent SPs practices, particularly among nurses compared to doctors. This result can be attributed to low educational levels (the majority of nurses did not have degrees, only a diploma) and lesser clinical experiences of participating nurses. Similarly, the study from Eskander et al. [26] rationalized noncompliance with SPs to limited staff numbers, increased workloads, and difficulty in maintaining hand hygiene. Zeb et al. [19] documented the contributing factors towards non-compliance of nurses to SPs were limited facilities, high workloads due to staff shortages, and inefficient policies and regulations for infection prevention.

In addition, the current study found that time constraints, uncomfortable PPE, patients who felt stigmatized when PPE was used, and a belief that infections cannot be acquired in hospital were perceived highly by doctors as challenges that affected HCWs’ adherence to SPs. This can be attributed to the lower attendance at training courses about SPs among doctors. Carvalho et al. [33] also found that 69% of doctors (in a sample of 58) stated that insufficient SP training and excess workload were the main reasons for their noncompliance with the SPs. Furthermore, Oliveira and Gonçalves [40] reported that reduced risk perception is the most significant causal factor for doctors’ noncompliance with SPs.

The current study found a significant correlation between overall compliance score with standard precautions and a nurse’s age, clinical experience, attendance of training courses, and need to attend training courses. In addition, the results illustrated no significant relationship between compliance score and gender, educational level, marital status, and nationality. This result agrees with Al-Faouri et al. [2] who documented a positive relationship between SPs compliance and HCWs’ increased knowledge, age, and duration of experience.

Haileet al. [8] also found that gender, knowledge about SPs, perception of infection risk, received training on SPs, the availability and accessibility of PPE, management support, workplace safety climate, and feedback on safety practices were significantly associated with compliance to SPs. Although their results contradicted the current study’s results regarding the association between gender and compliance to SPs, they found significant associations between both. Conversely, Zeb et al. [19] reported a significant correlation between compliance and gender (*p* = 0.006) and SP training (*p* = 0.001) among participating nurses.

Compliance with SPs among doctors is important in protecting them from transmission of infections, which usually result from invasive procedures and handling of body fluids or secretions. Results of a recent study showed that there was no significant relationship between overall compliance score with standard precautions and a doctor’s gender, age, educational level, marital status, nationality, clinical experience, attendance of training courses, as well as need to attend training courses. Conversely, Carvalho et al. [33], Akinade and Babatunde [41], and Luo et al. [42] found that greater work experience enhanced a doctor’s compliance with SPs, whereas age and marital status had lesser, insignificant relationships with compliance to SPs in those studies.

It is vital for healthcare organizations to offer regular training and necessary measures for all their nurses and other HCWs to enhance competence and capacity to SPs. Furthermore, it is recommended that healthcare organizations encourage their employees to raise awareness of basic standard precautions among their peers and to participate in policy development of basic SPs. Healthcare organizations, additionally, need to conduct regular meetings with all HCWs to discuss challenges or issues related to basic standard precautions practice and provide proper management strategies to foster a culture of safety and quality. It is also suggested that new training interventions on standard precautions practice be designed and evaluated to assess the long-term influence on HCWs. Ultimately, the general view is that healthcare practice must consider a holistic approach to promote adherence to SPs to enhance outcomes associated with optimized care delivery and stakeholder safety.

This cross-sectional study was limited to one university-affiliated hospital. Therefore, the results of this study cannot be generalized, as it is limited in terms of study participants, the setting, and the geographical area. The use of self-reported measures was another limitation of this study. With any self-reported measure, there is the potential for subjective bias. The use of more objective measures may provide more accurate results in future similar studies.

## 5. Conclusions

The current study illustrated that both nurses and doctors had poor compliance with almost all SPs (except disposal of sharps) in a teaching university hospital in KSA. Nurses’ compliance to recommended precautions varied significantly according to age, clinical experience, and attendance of training courses as well as need to attend training courses, so characteristics that influenced compliance were investigated. The results of the current study showed that excessive workload, absence of regular training on infection control, staff shortages, lack of hospital policies for practice of SPs, lack of adequate facilities and resources for practice of SPs, lack of a functional infection control committee, lack of a standardized documentation tool for SPs, lack of management support, and lack of knowledge of SPs were challenges that affected HCWs’ adherence to SPs practices, particularly among nurses compared to doctors.

## Figures and Tables

**Table 1 healthcare-11-02975-t001:** Comparison of the demographic data between doctors and nurses.

Demographic Data	Doctors (*n* = 49)		Nurses (*n* = 112)	
	f	%	f	%
Gender				
Male	43	87.8	16	14.3
Female	6	12.2	96	85.7
Age (years)				
<30	9	18.4	41	36.6
30–39	17	34.7	59	52.7
40–49	15	30.6	12	10.7
≥50	8	16.3	0	0.0
Mean ± SD.	39.71 ± 8.58		32.50 ± 5.58	
Educational level				
Diploma	0	0.0	64	57.1
Bachelor’s degree	4	8.2	47	42.0
Master’s degree	16	32.7	1	0.9
Doctoral degree	29	59.2	0	0.0
Marital status				
Single	7	14.3	39	34.8
Married	27	55.1	46	41.1
Divorced	12	24.5	20	17.9
Widowed	3	6.1	7	6.3
Nationality				
Saudi	30	61.2	39	34.8
Non-Saudi	19	38.8	73	65.2
Clinical experience (years)				
<5	7	14.3	17	15.2
5–14	19	38.8	78	69.6
15–19	9	18.4	10	8.9
≥20	14	28.6	7	6.3
Mean ± SD.	14.04 ± 7.82		9.71 ± 5.25	
Have you attended training courses in infection control?				
No	45	91.8	109	97.3
Yes	4	8.2	3	2.7
Do you need to enroll in training courses in infection control?				
Yes	27	55.1	66	58.9
No	22	44.9	46	41.1

**Table 2 healthcare-11-02975-t002:** Comparison of total scores of CSPS between the doctors and nurses.

Compliance with Standard Precautions Scale (CSPS)	Doctors (*n* = 49)	Nurses (*n* = 112)	t	*p*
	Mean ± SD	Mean ± SD		
Use of protective devices				
Total score	1.78 ± 1.10	1.86 ± 1.22	0.401	0.689
% score	29.59 ± 18.40	30.95 ± 20.36		
Disposal of sharps				
Total score	2.31 ± 0.58	2.22 ± 0.53	0.883	0.379
% score	76.87 ± 19.49	74.11 ± 17.73		
Disposal of waste				
Total score	0.06 ± 0.24	0.16 ± 0.37	2.026 *	0.045 *
% score	6.12 ± 24.22	16.07 ± 36.89		
Decontamination of spills and used items				
Total score	0.27 ± 0.67	0.46 ± 0.75	1.599	0.113
% score	8.84 ± 22.34	15.18 ± 24.87		
Prevention of cross infection from person to person				
Total score	2.24 ± 1.47	2.59 ± 1.39	1.421	0.157
% score	32.07 ± 20.93	36.99 ± 19.89		
Overall CSPS				
Total score	6.65 ± 3.15	7.29 ± 3.17	1.167	0.245
% score	33.27 ± 15.76	36.43 ± 15.85		

t: Student *t*-test; *p*: *p* value for comparing between the studied groups; *: statistically significant at *p* ≤ 0.05.

**Table 3 healthcare-11-02975-t003:** Comparison of levels of CSPS between the doctors and nurses.

Compliance with Standard Precautions Scale (CSPS)	Doctors (*n* = 49)		Nurses (*n* = 112)		t	*p*
	f	%	f	%		
Use of protective devices						
Poor	39	79.6	86	76.8	χ^2^	^MC^ *p* = 1.000
Suboptimal	9	18.4	22	19.6		
Satisfactory	0	0.0	2	1.8		
Optimal	1	2.0	2	1.8		
Disposal of sharps						
Poor	3	6.1	6	5.4	1.468	0.480
Suboptimal	28	57.1	75	67.0		
Optimal	18	36.7	31	27.7		
Disposal of waste						
Poor	46	93.9	94	83.9	2.975	0.085
Optimal	3	6.1	18	16.1		
Decontamination of spills and used items						
Poor	45	91.8	103	92.0	0.520	^MC^ *p* = 0.892
Suboptimal	3	6.1	5	4.5		
Optimal	1	2.0	4	3.6		
Prevention of cross infection from person to person						
Poor	43	87.8	86	76.8	7.266 *	^MC^ *p* = 0.041 *
Suboptimal	3	6.1	23	20.5		
Satisfactory	2	4.1	1	0.9		
Optimal	1	2.0	2	1.8		
Overall CSPS						
Poor	42	85.7	88	78.6	1.676	^MC^ *p* = 0.723
Suboptimal	6	12.2	21	18.8		
Satisfactory	0	0.0	1	0.9		
Optimal	1	2.0	2	1.8		

χ^2^: Chi-squared test; MC: Monte Carlo; *p*: *p* value for comparing between the studied groups; *: statistically significant at *p* ≤ 0.05.

**Table 4 healthcare-11-02975-t004:** Comparison between doctors and nurses of challenges affecting their adherence to standard precautions.

Q	Challenges Affecting HCWs in Their Adherence to SPs	Doctors (*n* = 49)				Nurses (*n* = 112)				χ^2^	*p*
		No		Yes		No		Yes			
		No.	%	No.	%	No.	%	No.	%		
1	Lack of knowledge of SPs	33	67.3	16	32.7	75	67.0	37	33.0	0.002	0.962
2	Belief that you will not acquire infection in the hospital	27	55.1	22	44.9	73	65.2	39	34.8	1.471	0.225
3	Lack of functional infection control committee	21	42.9	28	57.1	34	30.4	78	69.6	2.368	0.124
4	Absence of regular training on infection control	13	26.5	36	73.5	28	25.0	84	75.0	0.042	0.837
5	Lack of adequate facilities/resources for practicing SPs	22	44.9	27	55.1	32	28.6	80	71.4	4.076 *	0.043 *
6	Patients feel stigmatized when PPE is used	23	46.9	26	53.1	62	55.4	50	44.6	0.969	0.392
7	PPE is uncomfortable	14	28.6	35	71.4	46	41.1	66	58.9	2.278	0.131
8	Time constraints	10	20.4	39	79.6	32	28.6	80	71.4	1.178	0.278
9	Excess workload	12	24.5	37	75.5	20	17.9	92	82.1	0.942	0.332
10	Shortage of staff	17	34.7	32	65.3	29	25.9	83	74.1	1.294	0.255
11	Lack of management support	26	53.1	23	46.9	41	36.6	71	63.4	3.798	0.051
12	Lack of standardized documentation tool for SPs	20	40.8	29	59.2	37	33.0	75	67.0	0.902	0.342
13	Lack of hospital policy regarding SPs	21	42.9	28	57.1	30	26.8	82	73.2	4.068 *	0.044 *

χ^2^: Chi-squared test; *p*: *p* value for comparing between the studied groups; *: statistically significant at *p* ≤ 0.05; HCWs: healthcare workers; SPs: standard precautions; PPE: personal protective equipment.

**Table 5 healthcare-11-02975-t005:** Correlations between compliance with standard precautions and nurses’ demographic data.

	Compliance with Standard Precautions Scale (CSPS)					
	Use of PPE	Disposal of Sharps	Disposal of Waste	Decontamination of Spills and Used Items	Prevention of Cross Infection from Person to Person	Overall CSPS Score
Gender						
Male	31.25 ± 18.13	75.0 ± 19.25	18.75 ± 40.31	22.92 ± 31.55	34.82 ± 14.73	37.19 ± 15.49
Female	30.90 ± 20.80	73.96 ± 17.57	15.63 ± 36.50	13.89 ± 23.53	37.35 ± 20.66	36.30 ± 15.98
t (*p*)	0.063 (0.95)	0.217 (0.829)	0.312 (0.755)	1.349 (0.180)	0.469 (0.640)	0.206 (0.837)
Age (years)						
<30	28.86 ± 20.76	74.80 ± 16.30	24.39 ± 43.48	13.82 ± 28.84	33.80 ± 21.61	35.0 ± 18.77
30–39	29.94 ± 17.72	72.32 ± 18.73	8.47 ± 28.09	12.99 ± 20.55	35.11 ± 15.54	34.49 ± 10.74
40–49	43.06 ± 27.94	80.56 ± 17.16	25.0 ± 45.23	30.56 ± 26.43	57.14 ± 22.79	50.83 ± 19.75
F (*p*)	2.472 (0.089)	1.128 (0.328)	2.727 (0.070)	2.660 (0.074)	7.806 (0.001)	6.075 * (0.003 *)
Educational level						
Diploma	28.91 ± 17.88	71.88 ± 18.03	12.50 ± 33.33	11.46 ± 19.89	34.82 ± 18.51	33.98 ± 13.60
Bachelor’s degree	34.04 ± 23.30	77.30 ± 17.18	21.28 ± 41.37	20.57 ± 29.94	39.51 ± 21.52	39.79 ± 18.24
Master’s degree	16.67 ± 0.0	66.67 ± 0.0	0.0 ± 0.0	0.0 ± 0.0	57.14 ± 0.0	35.0 ± 0.0
F (*p*)	1.112 (0.332)	1.368 (0.259)	0.860 (0.426)	2.044 (0.134)	1.279 (0.283)	1.849 (0.162)
Marital status						
Single	28.63 ±20.57	73.50 ± 15.63	23.08 ± 42.68	16.24 ± 30.47	33.70 ± 21.36	35.0 ± 18.42
Married	33.33 ± 21.37	76.81 ± 18.42	10.87 ± 31.47	13.04 ± 22.75	38.20 ± 18.33	37.39 ± 15.34
Divorced	30.0 ± 17.61	73.33 ± 20.52	20.0 ± 41.04	18.33 ± 20.16	38.57 ± 19.72	37.25 ± 13.03
Widowed	30.95 ± 22.42	61.90 ± 12.60	0.0 ± 0.0	14.29 ± 17.82	42.86 ± 23.33	35.71 ± 13.05
F (*p*)	0.386 (0.763)	1.509 (0.216)	1.303 (0.277)	0.242 (0.867)	0.652 (0.584)	0.181 (0.909)
Nationality						
Saudi	32.91 ± 22.13	75.21 ± 18.29	15.38 ± 36.55	17.95 ± 27.41	37.36 ± 22.14	37.69 ± 17.47
Non–Saudi	29.91 ± 19.44	73.52 ± 17.53	16.44 ± 37.32	13.70 ± 23.46	36.79 ± 18.74	35.75 ± 14.99
t (*p*)	0.741 (0.461)	0.481 (0.631)	0.143 (0.886)	0.861 (0.391)	0.144 (0.885)	0.615 (0.540)
Clinical experience (years)						
<5	32.35 ± 25.32	76.47 ± 15.66	41.18 ± 50.73	21.57 ± 35.24	42.02 ± 26.47	41.18 ± 23.02
5–15	27.78 ± 17.13	73.50 ± 18.11	8.97 ± 28.77	11.11 ± 21.26	33.33 ± 16.06	33.14 ± 11.87
15–19	36.67 ± 18.92	70.0 ± 18.92	20.0 ± 42.16	23.33 ± 16.1	44.29 ± 25.6	41.5 ± 13.34
≥20	54.76 ± 28.41	80.95 ± 17.82	28.57 ± 48.8	33.33 ± 33.33	55.1 ± 20.91	54.29 ± 23.17
F (*p*)	4.50 * (0.005 *)	0.651 (0.584)	4.232 * (0.007 *)	2.802 * (0.043 *)	3.910 * (0.011 *)	5.537 * (0.011 *)
Have you attended training courses in infection control?						
Yes	31.19 ± 20.55	74.62 ± 17.51	16.51 ± 37.3	15.60 ± 25.08	37.61 ± 19.79	36.88 ± 15.82
No	22.22 ± 9.62	55.56 ± 19.25	0.0 ± 0.0	0.0 ± 0.0	14.29 ± 0.0	20.0 ± 0.0
t (*p*)	0.751 (0.454)	1.857 (0.066)	0.763 (0.447)	6.493 * (>0.001 *)	12.305 * (>0.001 *)	11.138 * (>0.001 *)
Do you need to enroll in training courses in infection control?						
Yes	34.09 ± 23.28	76.77 ± 16.51	19.70 ± 40.08	19.19 ± 28.68	39.83 ± 21.49	39.55 ± 18.22
No	26.45 ± 14.31	70.29 ± 18.89	10.87 ± 31.47	9.42 ± 16.72	32.92 ± 16.73	31.96 ± 10.25
t (*p*)	2.147 * (0.034 *)	1.925 (0.057)	1.303 (0.195)	2.269 * (0.025 *)	1.827 (0.070)	2.806 * (0.006 *)

* indicates statistically significant at *p* ≤ 0.05.

**Table 6 healthcare-11-02975-t006:** Correlations between compliance with standard precautions and doctors’ demographic data.

	Compliance with Standard Precautions Scale (CSPS)					
	Use PPE	Disposal Sharps	Disposal Waste	Decontamination of Spills Used Articles	Prevention of Cross Infection from Person to Person	Overall CSPS
Gender						
Male	28.29 ± 17.64	75.19 ± 19.37	4.65 ± 21.31	7.75 ± 21.62	30.23 ± 20.25	31.74 ± 14.92
Female	38.89 ± 22.77	88.89 ± 17.21	16.67 ± 40.82	16.67 ± 27.89	45.24 ± 22.89	44.17 ± 18.82
t (*p*)	1.331 (0.189)	1.641 (0.108)	0.708 (0.509)	0.914 (0.365)	1.676 (0.10)	1.854 (0.070)
Age (years)						
<30	35.19 ± 28.19	88.89 ± 16.67	11.11 ± 33.33	18.52 ± 37.68	39.68 ± 30.95	41.11 ± 25.95
30–39	29.41 ± 15.06	74.51 ± 18.74	5.88 ± 24.25	7.84 ± 18.74	36.97 ± 22.62	34.41 ± 13.91
40–49	25.56 ± 15.26	80.0 ± 16.90	0.0 ± 0.0	4.44 ± 11.73	24.76 ± 11.41	29.0 ± 8.28
≥50	31.25 ± 18.77	62.50 ± 21.36	12.50 ± 35.36	8.33 ± 23.57	26.79 ± 14.16	30.0 ± 14.88
F (*p*)	0.524 (0.668)	3.186 * (0.033 *)	0.617 (0.608)	0.758 (0.524)	1.537 (0.218)	1.275 (0.294)
Educational level						
Bachelor	45.83 ± 39.38	83.33 ± 19.25	25.0 ± 50.0	25.0 ± 50.0	50.0 ± 37.80	48.75 ± 36.60
Master	27.08 ± 27.08	77.08 ± 20.07	6.25 ± 25.0	4.17 ± 16.67	34.82 ± 22.72	32.81 ± 11.97
Doctoral	28.74 ± 17.19	75.86 ± 19.71	3.45 ± 18.57	9.20 ± 19.71	28.08 ± 15.96	31.38 ± 12.95
F (*p*)	1.795 (0.178)	0.252 (0.779)	1.416 (0.253)	1.426 (0.251)	2.243 (0.118)	2.256 (0.116)
Marital status						
Single	38.10 ± 31.50	80.95 ± 17.82	14.29 ± 37.80	23.81 ± 41.79	51.02 ± 28.40	45.71 ± 27.45
Married	27.78 ± 14.62	79.01 ± 20.98	3.70 ± 19.25	4.94 ± 15.20	28.57 ± 20.20	31.11 ± 12.73
Divorced	25.0 ± 13.30	75.0 ± 15.08	0.0 ± 0.0	5.56 ± 12.97	27.38 ± 12.86	29.17 ± 7.93
Widowed	44.44 ± 25.46	55.56 ± 19.25	33.33 ± 57.74	22.22 **±** 38.49	38.10 ± 16.50	40.00 ± 21.79
F (*p*)	1.536 (0.218)	1.488 (0.231)	1.988 (0.129)	1.864 (0.149)	2.709 (0.056)	2.237 (0.097)
Nationality						
Saudi	32.22 ± 19.04	78.89 ± 18.54	10.0 ± 30.51	10.0 ± 24.99	33.81 ± 22.66	35.33 ± 17.02
Non–Saudi	25.44 ± 17.0	73.68 ± 21.02	0.0 ± 0.0	7.02 ± 17.84	29.32 ± 18.12	30.0 ± 13.33
t (*p*)	1.265 (0.212)	0.909 (0.368)	1.795 (0.083)	0.452 (0.654)	0.727 (0.471)	1.158 (0.253)
Clinical experience (years)						
<5	40.48 ± 30.21	85.71 ± 17.82	14.29 ± 37.80	23.81 ± 41.79	46.94 ± 31.64	45.71 ± 28.05
5–14	28.07 ± 14.75	77.19 ± 19.41	5.26 ± 22.94	7.02 ± 17.84	34.59 ± 22.50	33.42 ± 13.44
15–19	25.93 ± 14.70	81.48 ± 17.57	0.0 ± 0.0	3.70 ± 11.11	22.22 ± 10.38	28.33 ± 6.12
≥20	28.57 ± 17.82	69.05 ± 20.52	7.14 ± 26.73	7.14 ± 19.30	27.55 ± 13.10	30.0 ± 13.01
F (*p*)	0.992 (0.405)	1.441 (0.243)	0.457 (0.714)	1.300 (0.286)	2.329 (0.087)	2.082 (0.116)
Have you attended training courses in infection control?						
Yes	28.52 ± 15.74	76.30 ± 19.62	2.22 ± 14.91	7.41 ± 18.65	29.84 ± 17.18	31.67 ± 12.06
No	41.67 ± 39.67	83.33 ± 19.25	50.0 ± 57.74	25.0 ± 50.0	57.14 ± 42.06	51.25 ± 37.05
t (*p*)	0.658 (0.556)	0.688 (0.495)	1.650 (0.196)	0.699 (0.534)	1.289 (0.285)	1.052 (0.369)
Do you need to enroll in training courses in infection control?						
Yes	30.86 ± 22.03	75.31 ± 21.86	7.41 ± 26.69	8.64 ± 23.74	36.51 ± 24.85	35.0 ± 19.01
No	28.03 ± 13.0	78.79 ± 16.41	4.55 ± 21.32	9.09 ± 21.04	26.62 ± 13.44	31.14 ± 10.57
t (*p*)	0.560 (0.579)	0.618 (0.540)	0.408 (0.685)	0.069 (0.945)	1.773 (0.084)	0.851 (0.399)

* indicates statistically significant at *p* ≤ 0.05.

## Data Availability

The data that support the findings of this study are available from the corresponding author upon reasonable request.

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
