# Peer review of "Compliance among Registered Nurses and Doctors in Critical Care Units: Challenges Affecting Their Adherence to Standard Precautions"

_healthcare, 2023, doi:10.3390/healthcare11222975_

Round 1
Reviewer 1 Report
Comments and Suggestions for Authors
Thank you for the opportunity to review manuscript healthcare-2682271 titled "Compliance among registered nurses and doctors in critical care units: Challenges affecting their adherence to standard precautions." Overall this is a well written manuscript that addresses an important topic. I believe this paper will be of interest to a wide audience.
I have a few comments/editorial suggestions to consider before accepting this for publication.
1.Abstract
· It would be helpful to identify how data was collected within the abstract.
· Conclusion - …except disposal of sharps, with suboptimal compliance – is there a word missing?
· Should offer training (remove sessions) in the last sentences. Training can take many forms and it does not always involve “sessions.”
2. Introduction
· Page 2 line 49 - suggest replacing “is” with applies
· Line 51-52 – consider adding spreading -noncompliance increase the “risk of contracting and spreading”
· Line 79 – Suggesting replacing “Their” with the authors name since it is not written out in the preceding sentence.
· There are several small paragraphs that discuss previous studies conducted in the area. Suggest combing these into a single paragraph to allow for a more cohesive presentation of the literature.
3. Materials and Methods
· It would be important to know when the study was conducted given the emphasis placed on infection control and standard precautions during the recent pandemic.
· Page 3 line 104 – typo should be consist of (not qunsist)
· Page 3 line 108-109 – martial status is listed twice – remove one.
· Page 3 line 125. Was the tool translated to Arabic? If so, this should be made explicit.
· Data Collection – suggest rearranging so the process is presented in chronological order. It is assumed the consent process (lines 135-137) occurred prior to delivering the surveys (line 131-132)?
4. Results
· Well done and easy to follow. The tables are particularly well done and are very effective at presenting the data in a cohesive manner.
· Page 8 line 250, suggest replacing done with completed.
· Page 10 line 273 there appears to be something missing with “and each of doctors”
5. Discussion
· It would be helpful if the authors could offer a few recommendations. Specific interim of training/retraining, future research etc.
· The use of self-report surveys is always a limitation. This should be mentioned with the study limitations.
Author Response
Dear Reviewer,
Thank you for your time editing our article. All your comments and suggestions to improve the manuscript have been considered and the article is edited accordingly.
1.Abstract
- It would be helpful to identify how data was collected within the abstract.
Added: The researchers administered the compliance with the Standard Precautions Scale (CAPS) to staff nurses and doctors from critical care units at a university-
- Conclusion - …except disposal of sharps, with suboptimal compliance – is there a word missing?
Edited: The study found that both nurses and doctors had poor compliance with all SPs except for disposal of sharps, which scored with suboptimal compliance
- Should offer training (remove sessions) in the last sentences. Training can take many forms and it does not always involve “sessions.”
Edited: organizations should consider a holistic approach to promote adherence to SPs to enhance outcomes associated with optimized care delivery and stakeholder safety.
2. Introduction
4. Page 2 line 49 - suggest replacing “is” with applies
Modified, now line 55
- Line 51-52 – consider adding spreading -noncompliance increase the “risk of contracting and spreading”
Modified now line 67-68
- Line 79 – Suggesting replacing “Their” with the authors name since it is not written out in the preceding sentence.
Modified
- There are several small paragraphs that discuss previous studies conducted in the area. Suggest combing these into a single paragraph to allow for a more cohesive presentation of the literature.
Modified. Please take a look at the intro now.
3. Materials and Methods
8. It would be important to know when the study was conducted given the emphasis placed on infection control and standard precautions during the recent pandemic.
Data was collected from 1st of February to May 2019. Line 167
- Page 3 line 104 – typo should be consist of (not qunsist)
edited
- Page 3 line 108-109 – martial status is listed twice – remove one.
edited
- Page 3 line 125. Was the tool translated to Arabic? If so, this should be made explicit.
No, it was not, sorry for the confusion.
- Data Collection – suggest rearranging so the process is presented in chronological order. It is assumed the consent process (lines 135-137) occurred prior to delivering the surveys (line 131-132)?
Kindly check the method section now.
4. Results
13. Well done and easy to follow. The tables are particularly well done and are very effective at presenting the data in a cohesive manner.
Thank you
- Page 8 line 250, suggest replacing done with completed.
edited
- Page 10 line 273 there appears to be something missing with “and each of doctors”
The phrase was edited.
5. Discussion
16. It would be helpful if the authors could offer a few recommendations. Specific interim of training/retraining, future research etc.
Recommendations added line 429-440
- The use of self-report surveys is always a limitation. This should be mentioned with the study limitations.
This limitation was added.
Reviewer 2 Report
Comments and Suggestions for Authors
The summary is precise, clear, and concise. Gives a global overview of the article proposal.
I propose that you reduce the keywords to five in total. It is more or less standard, not to exceed five keywords.
On line 60, the sentence must be rewritten so that there is only one quote per sentence. In this case, two authors appear in the same sentence: 7 and 8. In line 76, it is correct, the references appear at the end of the sentence. They are different cases. I hope I was clear with these two examples.
The introduction, corresponding to the literature review, is supported… in each sentence there is almost always an author, this section should be expanded further. The broader the literature review, the more in-depth the conclusions will be. For this reason, I propose that this section be revised to expand.
Regarding the questionnaire, I got the impression that they had adapted another questionnaire…. Right?! It is important to highlight the fact that the questionnaire you administered was validated… Was it validated? This aspect is crucial because if it has not been properly validated, it could jeopardize the data collected and its respective analysis. Attention: Authors must send evidence that proves that the questionnaire was validated. Otherwise, the article must be rejected.
When analyzing data, in addition to using statistical methods, it is not enough to put into text what is in the table. It is necessary to go further... the authors need to relate the data with those that preceded it, it is necessary to be more reflective and critical... what I read was just a description, which is not enough. In this section, for all data, at the end of each item, there must be at least one critical-reflective sentence.
The Discussion section seems to be doing well... it is now more critical and reflective and includes authors, which allows us to support the analysis they carried out.
The conclusions should be more expanded… the connection with the literature review. They should present more proposals for the future… they seem to be very summarized. This part must be changed, which means it must be expanded.
The bibliography presented is in line with the issue, is relatively current and diverse, but should be further expanded.
Author Response
Dear Reviewer,
Thank you for your time editing our article. All your comments and suggestions to improve the manuscript have been considered and the article is edited accordingly.
- The summary is precise, clear, and concise. Gives a global overview of the article proposal.
Thank you
- I propose that you reduce the keywords to five in total. It is more or less standard, not to exceed five keywords.
Keywords were reduced
- On line 60, the sentence must be rewritten so that there is only one quote per sentence. In this case, two authors appear in the same sentence: 7 and 8. In line 76, it is correct, the references appear at the end of the sentence. They are different cases. I hope I was clear with these two examples.
Kindly check the intro again, it was modified for enhancement
- The introduction, corresponding to the literature review, is supported… in each sentence there is almost always an author, this section should be expanded further. The broader the literature review, the more in-depth the conclusions will be. For this reason, I propose that this section be revised to expand.
Kindly check the intro again, it was modified and extended.
- Regarding the questionnaire, I got the impression that they had adapted another questionnaire…. Right?! It is important to highlight the fact that the questionnaire you administered was validated… Was it validated? This aspect is crucial because if it has not been properly validated, it could jeopardize the data collected and its respective analysis. Attention: Authors must send evidence that proves that the questionnaire was validated. Otherwise, the article must be rejected.
Apologize for the confusion. The tools were utilized as it is, not adopted.
- When analyzing data, in addition to using statistical methods, it is not enough to put into text what is in the table. It is necessary to go further... the authors need to relate the data with those that preceded it, it is necessary to be more reflective and critical... what I read was just a description, which is not enough. In this section, for all data, at the end of each item, there must be at least one critical-reflective sentence.
Kindly check the results section again, it was modified.
- The Discussion section seems to be doing well... it is now more critical and reflective and includes authors, which allows us to support the analysis they carried out.
Thank you
- The conclusions should be more expanded… the connection with the literature review. They should present more proposals for the future… they seem to be very summarized. This part must be changed, which means it must be expanded.
Other reviewer suggested to shorten the conclusion. Hence, conclusion was modified to fulfill each one of the reviewers’ comments including this comment. Recommendations were added in the discussion section line 429-440
- The bibliography presented is in line with the issue, is relatively current and diverse, but should be further expanded.
Bibliography was expanded.
Reviewer 3 Report
Comments and Suggestions for Authors
This research show a cross-sectional descriptive study of compliance among registered nurses and doctors in critical care units. The challenges affecting their adherence to standard precautions are a topic necessary to study in Arab countries. I consider the topic relevant, but i think that the authors should be some changes before, to be clear some lack of the paper.
The introduction section is adequate. The objective must be written in past not future verb tense (lines 89-91). And the Materials and Methods section is adequate. The study design is correct and controls all study variables.
The result section: Tables (1-6) must be presented according to the journal's standards.
Conclusion must responde to the main aim of the study. It is advisable that conclusion be clear and concise. Therefore, it is not recommended that it is lenght be so long. 3-5 line.
The references must be reviewed completely. See the Reference List and Citations Guide. For example: Kim, H.; Hwang, Y.H. Factors contributing to clinical nurse compliance with infection prevention and control practices: A 448 cross-sectional study. Nurs Health Sci. 2020, 22, 126-133.
Institutional Review Board Statement section should be indicated the number of code.
Thank you for this helpful contribution. I applaud the effort of promoting these studies.
Author Response
Dear Reviewer,
Thank you for your time editing our article. All your comments and suggestions to improve the manuscript have been considered and the article is edited accordingly.
- The objective must be written in past not future verb tense (lines 89-91). And the Materials and Methods section is adequate. The study design is correct and controls all study variables.
Edited. Thank you for your comment.
- The result section: Tables (1-6) must be presented according to the journal's standards.
Edited
- Conclusion must responde to the main aim of the study. It is advisable that conclusion be clear and concise. Therefore, it is not recommended that it is lenght be so long. 3-5 line.
Edited.
- The references must be reviewed completely. See the Reference List and Citations Guide. For example: Kim, H.; Hwang, Y.H. Factors contributing to clinical nurse compliance with infection prevention and control practices: A 448 cross-sectional study. Nurs Health Sci. 2020, 22, 126-133.
edited
- Institutional Review Board Statement section should be indicated the number of code.
The number code was added (Ref No 2F.76).
Reviewer 4 Report
Comments and Suggestions for Authors
Dear Editor,
Thank you for giving me the opportunity to review this manuscript. This manuscript is well-written. Overall, the messages are conveyed very well. I just have a couple of comments to enhance the clarity of this manuscript.
1. Introduction: Page 2 line 58, authors wrote “In numerous studies”, but cited only one. Please provide more than one citation if you stated “numerous studies”.
2. Materials and methods: How did you decide the number of samples as 161 participants? Please explain how you came with this number of participants.
3. Page 3 line 104, the word “qunsist”. Is this a typo? Please correct the word.
4. For the two questionnaires that you adopted from others, please make sure that you obtained permission from the legal owners to use their questionnaires in your study.
5. Since you adopted the questionnaires from other countries, what language did you use when you collect the data in the questionnaire (because for CSPS, you wrote “after translation”)? Translation to which language? Also, for the questionnaire by Ogoina et al, did you use English to your participants or did you translate it to another language well? In case you translated, how was the translation process? Please explain.
6. Page 3 line 128, the value of 0875. Please check what is missing in this value.
7. Page 4 table 1 and page 6 table 3, please change “No.” to “f”, because the value you give for each item is the frequency, not number.
8. Page 14 line 443, acknowledgement. Please revise this part.
Thank you and good luck for the revision.
Author Response
Dear Reviewer,
Thank you for your time editing our article. All your comments and suggestions to improve the manuscript have been considered and the article is edited accordingly.
- Materials and methods: How did you decide the number of samples as 161 participants? Please explain how you came with this number of participants.
Sampling section has been added :
Sampling
Convenience (nonprobability) sampling was used to obtain the study sample. Convenience sampling was chosen due to the descriptive nature of the study. A convenience sample of 161 participants comprising nurses (n =112) and doctors (n=49) were recruited. Based on the extant literature on infection control and prevention standards [22], using the following statistical parameters: medium effect size of .30, alpha of .10, and power of .80, a total of 161 subjects is sufficient to ensure statistically and clinically meaningful results.
- .Page 3 line 104, the word “qunsist”. Is this a typo? Please correct the word.
Edited. Thank you
- For the two questionnaires that you adopted from others, please make sure that you obtained permission from the legal owners to use their questionnaires in your study.
The tools was utilized not adopted, apologies for the confusion, permission was granted.
- Since you adopted the questionnaires from other countries, what language did you use when you collect the data in the questionnaire (because for CSPS, you wrote “after translation”)? Translation to which language? Also, for the questionnaire by Ogoina et al, did you use English to your participants or did you translate it to another language well? In case you translated, how was the translation process? Please explain.
The tools was utilized not adopted, apologies for the confusion. They were also used in English.
- Page 3 line 128, the value of 0875. Please check what is missing in this value.
Corrected .08
- Page 4 table 1 and page 6 table 3, please change “No.” to “f”, because the value you give for each item is the frequency, not number.
Changed, thank you
- Page 14 line 443, acknowledgement. Please revise this part.
Revised. Thank you